

# The aridity index governs the variation of vegetation characteristics in alpine grassland, Northern Tibet Plateau

Biying Liu[1,2], Jian Sun[2,3], Miao Liu[2], Tao Zeng[1] and Juntao Zhu[2]

[1] College of Earth Sciences, Chengdu University of Technology, Chengdu, China
[2] Synthesis Research Centre of Chinese Ecosystem Research Network, Key Laboratory of Ecosystem Network Observation and Modelling, Institute of Geographic Sciences and Natural Resources Research, Chinese Academy of Sciences, Beijing, China
[3] State Key Laboratory of Urban and Regional Ecology, Research Center for Eco-environmental Sciences, Chinese Academy of Sciences, Beijing, China

## ABSTRACT

The vegetation dynamic (e.g., community productivity) is an important index used to evaluate the ecosystem function of grassland ecosystem. However, the critical factors that affect vegetation biomass are disputed continuously, and most of the debates focus on mean annual precipitation (MAP) or temperature (MAT). This article integrated these two factors, used the aridity index (AI) to describe the dynamics of MAP and MAT, and tested the hypothesis that vegetation traits are influenced primarily by the AI. We sampled 275 plots at 55 sites (five plots at each site, including alpine steppe and meadow) across an alpine grassland of the northern Tibet Plateau, used correlation analysis and redundancy analysis (RDA) to explore which key factors determine the biomass dynamic, and explained the mechanism by which they affect the vegetation biomass in different vegetation types via structural equation modelling (SEM). The results supported our hypothesis, in all of the environmental factors collected, the AI made the greatest contribution to biomass variations in RDA , and the correlation between the AI and biomass was the largest ($R = 0.85$, $p < 0.05$). The final SEM also validated our hypothesis that the AI explained 79.3% and 84.4% of the biomass variations in the alpine steppe and the meadow, respectively. Furthermore, we found that the soils with higher carbon to nitrogen ratio and soil total nitrogen had larger biomass, whereas soil organic carbon had a negative effect on biomass in alpine steppe; however, opposite effects of soil factors on biomass were observed in an alpine meadow. The findings demonstrated that the AI was the most critical factor affecting biomass in the alpine grasslands, and different reaction mechanisms of biomass response to the AI existed in the alpine steppe and alpine meadow.

Corresponding authors
Jian Sun, sunjian@igsnrr.ac.cn
Tao Zeng, zengtao@cdut.cn

## INTRODUCTION

Grasslands are distributed widely in China and cover a total area of 4 million km$^2$ (*Sun, Ma & Lu, 2017*). As one of the most widespread vegetation types and an important component of the terrestrial ecosystem (*Li et al., 2016*), grasslands occupy approximately 1.5 million

km$^2$ of the Tibetan Plateau. Additionally, alpine grasslands (alpine steppe and meadow) dominate the natural vegetation of the Tibetan Plateau and cover more than 60% of its area (*Yang et al., 2009*), and alpine grassland plays an essential role in maintaining the ecosystem functions of the Tibetan Plateau. However, due to the harsh climate of the Tibetan Plateau, the environmental factors under the equilibrium conditions of the fragile surface system are usually in the critical threshold state (*Sun et al., 2019*). Small fluctuations of climate change will also generate strong responses in the ecosystem (*Zhang et al., 2017*) and lead to changes in the pattern, process and function of the plateau grassland ecosystems; the manifestations of this including grassland degradation and disappearance (*Li et al., 2013*; *Yu et al., 2012*). Consequently, the response mechanism of alpine grasslands to soil properties, climate and other environmental factors is a new scientific problem facing the research on environmental change of the Tibetan Plateau.

Biotic and abiotic factors constitute the fundamental forces that drive the quantity, distribution, structure, and diversity of vegetation communities. Numerous studies have investigated the effects of precipitation, temperature, different land use types, and human disturbance on soil properties, biodiversity, and biomass in grassland ecosystems (*Redmann et al., 1993*; *Sala et al., 1988*; *Sun, Cheng & Li, 2013*). They found that increased variability in precipitation had a strong effect on some arid land ecological processes (*Thomey et al., 2015*). Similarly, precipitation influenced aboveground biomass (BIO) and primary productivity strongly in most alpine grasslands (*Hu et al., 2010*; *Huxman et al., 2004*; *O'Connor, Haines & Snyman, 2001*; *Yang et al., 2009*). *Adler & Levine (2007)* hold that with increased mean annual precipitation (MAP) across a broad geographic gradient the mean plant species richness increased significantly. Additionally, temperature is another a critical factor, and some simulated warming experiments in alpine grasslands have shown that warming was associated with a decreased trend in aboveground net primary productivity and plant diversity (*Cantarel, Bloor & Soussana, 2013*; *White, Bork & Cahill, 2014*); meanwhile, a negative correlation was observed between BIO and temperature when annual precipitation was held constant at about 50 mm (*Epstein et al., 1996*; *Sala et al., 1988*). It is important to note that responses of grassland to climate change also varied if soil properties are different (*Noymeir, 1973*), for example, by increasing soil temperature reduced total plant community coverage in the peak growing peak season (*Wang et al., 2015*). Further, soil nitrogen also influenced species richness in a wide range of temperate grasslands (*Critchley et al., 2010*). Previous studies have indicated that plant-soil interactions were vital mechanisms in a grassland ecosystem (*Orwin et al., 2010*). Nevertheless, it is still controversial whether precipitation or temperature is the primary limiting environmental factor for vegetative traits (*Sun, Qin & Yang, 2016*; *Sun, Cheng & Li, 2013*; *Sun & Qin, 2016*; *Wu et al., 2013*). A limited number of studies has combined water and heat availability to address the contribution of climatic factors on vegetation characteristics. This study integrated these environmental variables, using the aridity index (AI) to describe the dynamics of precipitation and temperature. This study also proposed the hypothesis that the vegetation traits are influenced primarily by the AI. Because of the significant differences in precipitation and temperature between different vegetation types, we divided the research objects into steppe and meadow, then compared the contributions

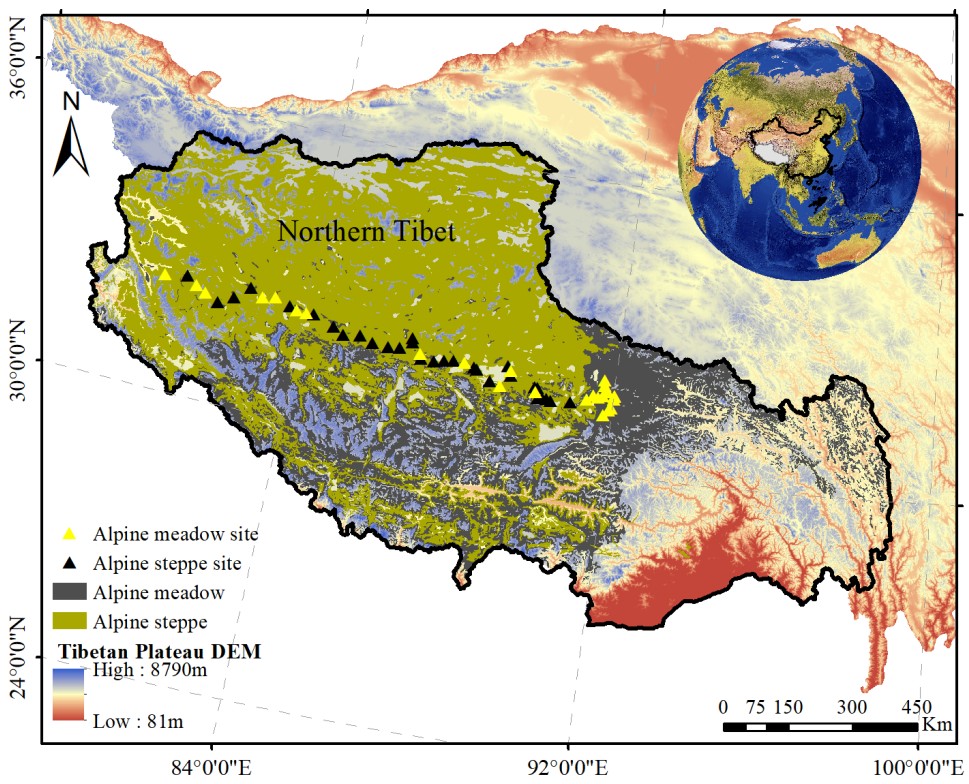

**Figure 1** **The distribution of sampled sites in alpine grasslands across northern Tibet Plateau.** The yellow solid triangles represent the samples collected in alpine meadow while the black solid triangles represent the samples collected in alpine steppe. In addition, the gray areas and the olive areas represent alpine meadow and steppe, respectively.

of the critical environmental variables to vegetation biomass, and explained the mechanism of environmental variables affecting biomass in these two vegetation types respectively.

## MATERIALS & METHODS

### Study area

The study was conducted on the alpine grassland in the northern Tibet Plateau (80.38°–92.01°E, 31.22°–32.51°N, average altitude of ∼4,600 m) of southwestern China (Fig. 1). The typical continental plateau climate is characterized by low temperature and limited precipitation, with a short, cool summer and long, cold winter. The annual precipitation ranges from 79.91 to 465.27 mm, and the mean annual temperature is 1.35°. Alpine grassland is one of the main ecosystems in the Tibetan Plateau, where the vegetation is alpine meadow (dominated by *Kobresia spp. and Poa spp.*) and alpine steppe (dominated by *Stipa purpurea*) (*Sun et al., 2009*; *Zeng, Wu & Zhang, 2015*). Alpine meadow has a sub-frigid and semi-arid high-plateau monsoon climate, and alpine steppe has a frigid and high-plateau dry climate. Alpine meadow soil and alpine steppe soil are very compact and loosely structured, respectively (*Zeng et al., 2018*).

## Survey design and sampling

Plant BIO were collected from 55 sites (30 sites in alpine steppe and 25 sites in alpine meadow) along a transect at spatial intervals of 50 km in August 2012 (Fig. 1, Table 1). Five plots (replicates) of 1 m × 1 m were randomly assigned to each sampling site (5 m² area). Plant BIO in these quadrats was harvested by a clipping method flush with the ground. All species information, including the vegetation coverage within each quadrat, was identified by a field guide (*Cai, Huang & Lang, 1989*), and the number of species in each quadrat was recorded to estimate diversity (*Whittaker, 1972*). After the data were aggregated, all of the plant samples were de-enzymed at 85 °C for 30 min and oven-dried at 65 °C until a constant weight was achieved. Thereafter, the samples were weighed on an electronic scale (accurate to one one-hundredth of a gram).

Soil from the 0–30 cm layer was also sampled in the plots, and soil samples were stored in a refrigerator at 4 °C until measuring chemical properties. We collected five soil cores using a 5 cm diameter soil auger and mixed them *in situ* into one composite sample. The soil organic carbon (SOC) and soil total nitrogen (STN) concentrations were determined using a vario MACRO cube elemental analyser (*Ganjurjav et al., 2016*). The mean annual temperature (MAT) and MAP were the mean values of temperature and precipitation in 2012 obtained from the Worldclim database (URL: http://www.worldclim.org/), and the outliers of the meteorological data acquired has been modified or removed. The database mentioned includes information on current global climate based on data collected from 1950 to 2000 and interpolated to a resolution grid of 2.5 arc min (∼5 km² grid cells at the equator, and smaller cells elsewhere) using information on latitude, longitude, and elevation (*Hijmans et al., 2005*). The AI was calculated using the MAT and MAP data according to Martonne's formula (*De Martonne, 1926*).

$$AI = \frac{P}{T + 10} \qquad (1)$$

where P is MAP (mm) and T is MAT (°C).

## Data analysis

The data package required in this study includes the target variable of BIO (BIO), and biotic factors [coverage (COV), and species richness (SR)] and abiotic factors [(MAP, MAT, the AI, STN, SOC), and the soil carbon to nitrogen ratio (C: N)] were selected to explore the response of BIO to these environment factors. Table 1 shows the specific information of the database.

First, we conducted a Pearson's correlation analysis (Corrplot package) using R software 2.11.1 (*R Core Team, 2018*) to screen out the variables that can reflect biomass changes. Second, the redundancy analysis is a linear canonical ordination method that is related closely to the potential explanatory variables, and it is effective in evaluating the relations among multiple interacting variables (*Tang et al., 2017*). Thus, R (Vegan) was employed to explore the relationships between plant biomass and the related variables that were filtered out and identify the key factor influencing BIO. To verify that the AI produces a variation in biomass, significant effects of plant biomass and the AI between the two types of vegetation were identified with an analysis of variance (ANOVA) in the R package (Psych).

Liu et al. (2019), *PeerJ*, DOI 10.7717/peerj.7272

**Table 1  Description of the alpine grassland sampled sites across northern Tibet Plateau.**

| Alpine ecosystem | Descriptive statistics | Longitude (°) | Latitude (°) | Altitude (m) | SOC (%) | STN (%) | C/N | MAT (°C) | MAP (mm) | AI | COV (%) | SR | BIO (g m$^{-2}$) |
|---|---|---|---|---|---|---|---|---|---|---|---|---|---|
| | Maximum | 90.95 | 32.51 | 4953 | 4.69 | 0.15 | 54.31 | 3.56 | 436.50 | 40.45 | 57.4 | 11.6 | 101.6 |
| | Minimum | 80.95 | 31.36 | 4406 | 1.08 | 0.05 | 10.31 | −0.07 | 100.96 | 7.98 | 1.9 | 2 | 1.1 |
| Alpine steppe ($n = 30$) | Mean | 86.99 | 31.88 | 4624 | 2.23 | 0.09 | 28.94 | 1.80 | 299.03 | 25.74 | 26.6 | 6.4 | 32.4 |
| | Median | 87.25 | 31.89 | 4600 | 2.12 | 0.09 | 26.48 | 2.15 | 343.37 | 27.22 | 26.9 | 6.2 | 22.27 |
| | Maximum | 92.01 | 32.43 | 4814 | 7.72 | 0.47 | 101.26 | 3.44 | 465.27 | 48.53 | 82.4 | 11.6 | 149.3 |
| Alpine meadow ($n = 25$) | Minimum | 80.38 | 31.22 | 4374 | 0.26 | 0.04 | 6.91 | −0.70 | 79.91 | 6.98 | 2.4 | 1.8 | 3.2 |
| | Mean | 88.56 | 31.82 | 4584 | 2.37 | 0.18 | 27.82 | 0.80 | 351.03 | 33.97 | 43.7 | 6.9 | 59.4 |
| | Median | 91.33 | 31.72 | 4586 | 2.17 | 0.16 | 11.83 | 0.16 | 451.42 | 43.79 | 49 | 7.4 | 56.90 |

**Notes.**

AI, SOC, C: N, STN, SR, BIO, and COV represent aridity index, soil organic carbon, the soil carbon to nitrogen ratio, soil total nitrogen, species richness, aboveground biomass, and coverage, respectively.

To demonstrate the reliability of our conclusions, we only used the AI and remaining independent variables collected to explore the mechanism of biomass variation. The structural equation modelling (SEM) approach was adopted to explain the mechanism by which the key factors affect BIO, both directly and indirectly, in alpine steppe and meadow. SEM is a common multivariate technique that has been used in recent studies to evaluate explicitly the causal relations among multiple interacting variables (*Sun, Ma & Lu, 2017*). SEM differs from other modeling approaches as it tests both the direct and indirect effects on pre-assumed causal relationships (*Fan et al., 2016*). The standard estimate results express the influence on the BIO using a path coefficient generated by Amos software (17.0.2; Amos Development Corporation, Crawfordville, FL, USA).

## RESULTS

### The relationships between environmental factors and aboveground biomass across alpine grasslands

According to Fig. 2, BIO was significantly correlated with four environmental variables at the 95% confidence interval level, including MAT ($R = -0.76$; $p < 0.05$), MAP ($R = 0.76$; $p < 0.05$), C: N ($R = -0.52$; $p < 0.05$), and the AI ($R = 0.85$; $p < 0.05$) in alpine grasslands. In addition, these environmental factors above were also closely related to each other, example for AI, it was significantly correlated with MAP ($R = -0.74$; $p < 0.05$) and MAT ($R = 0.97$; $p < 0.05$).

### Relative importance of the environmental factors

The results showed that the interactions among MAP, MAT, C: N, and the AI were the main drivers of the response variable (Fig. 3), explaining 79.82% of the variation in BIO. Specifically, the AI had the greatest influence, explaining 72.16% of the BIO; followed by MAP, which explained 58.34%; and MAT and C: N, which explained 57.34% and 26.25% of the BIO, respectively.

### The differences of aboveground biomass and the AI in the different grassland types

The BIO and the AI exhibited large differences between alpine steppe and meadow (Fig. 4 and Table 1). The BIO ranged from 1.14 to 101.65 g m$^{-2}$ in steppe, with a mean of 32.4 gm$^{-2}$, and 3.18 to 149.30 g m$^{-2}$ in meadow, with a mean of 59.4 g m$^{-2}$ (Table 1); the BIO of the latter was significantly higher than that of steppe ($p < 0.01$, Fig. 4B). The AI between steppe and meadow was also entirely different ($p < 0.05$, Fig. 4A). As shown in Table 1, the AI of alpine meadow largely was greater than 43.79, whereas it was less than 27.22 in alpine steppe. The regression analysis demonstrated that the BIO was exponential increasing with the AI values in both alpine steppe ($R^2 = 0.75$; $p < 0.001$; Fig. 5) and meadow ($R^2 = 0.77$; $p < 0.001$; Fig. 5). Notably, the vegetation in the alpine steppe and meadow exhibited different response rates to the AI. For BIO, with the same increase of AI, larger increases would be found in steppe than that in meadow, and when the AI value is around 26, the BIO of the two tend to be equal.

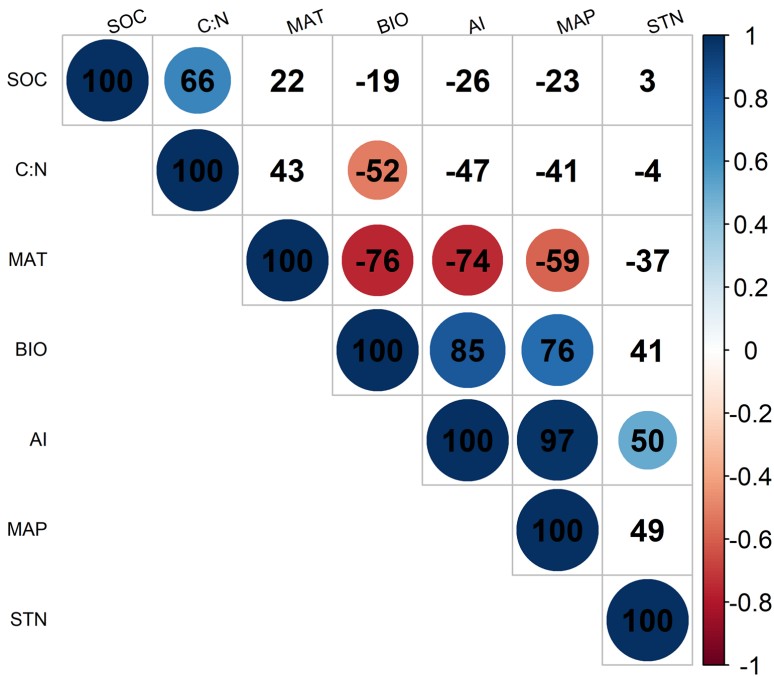

**Figure 2 The correlationships of aboveground biomass with environmental factors.** The colored solid circles represent the significant correlation ($p < 0.05$). And the MAT, MAP, AI, SOC, C: N, STN and BIO represent mean annual temperature, mean annual precipitation, aridity index, soil organic carbon, the soil carbon to nitrogen ratio, soil total nitrogen and aboveground biomass, respectively.

## Using structural equation modelling to explore the variations in aboveground biomass of alpine meadow and steppe

The SEM results explained 84.1% (Fig. 6A) and 87.7% (Fig. 6B) of the variation in the BIO in alpine steppe and meadow, respectively. Table 2 shows a summary of the direct, indirect, and total effects of the variables. Clearly, in steppe and meadow, all independent variables included environmental factors (SOC, STN, C: N, and the AI) and biotic factors (coverage and species richness), both of which affected the dynamic change of BIO, and all of the variables above are regulated directly or indirectly by the AI. As the SEM model results show (Table 2, Fig. 6), we also verified that the AI is the most critical factor affecting the variation in BIO. Furthermore, the weights contributed up to 79.3% (path coefficient = 0.793) of the variation in BIO in the alpine steppe (Fig. 6A) and 84.4% (path coefficient = 0.844) to the variation in BIO in alpine meadow (Fig. 6B).

In addition, the SEM showed that C: N and STN had positive effects while SOC had negative effects on BIO in alpine steppe, with path coefficients of 0.331, 0.078, and −0.091, respectively (Fig. 6A). However, in alpine meadow, we observed the opposite effect, as C: N and STN had negative effects on BIO while SOC had positive effects. Their path coefficients were −0.303, −0.161, and 0.111, respectively (Fig. 6B).

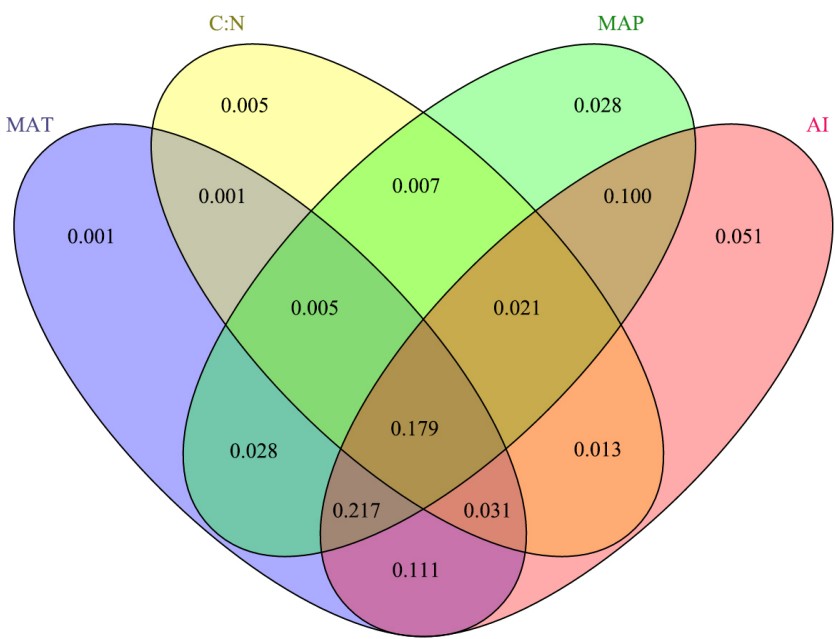

**Figure 3** **The contributions (%) of the different critical environmental variables to aboveground biomass via redundancy analysis.** The MAT, C: N, MAP and AI represent mean annual temperature, the soil ratio of carbon to nitrogen, mean annual precipitation and the aridity index, respectively.

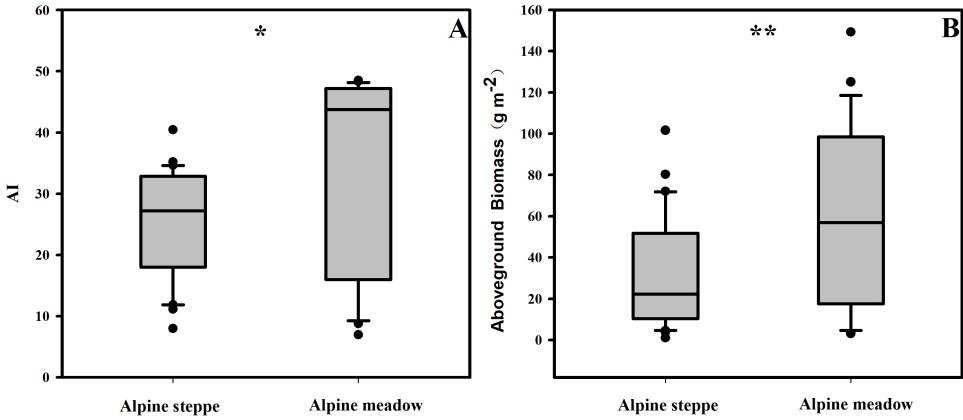

**Figure 4** **The variances of AI (A) and aboveground biomass (B) in alpine steppe and meadow, respectively.** ** and * represent the significance at the 0.01 and 0.05 level.

## DISCUSSION

Our results demonstrated that the AI was the primary limiting environmental factor for vegetation traits (Figs. 2, 3 and 5), and this finding supported our hypothesis. As we know, precipitation and temperature are key factors that affect the vegetation dynamic, and the aboveground biomass/vegetation index increased or decreased significantly depending on the more or less precipitation (*Sun et al., 2013*; *Sun & Qin, 2016*; *Sun, Qin & Yang, 2016*)

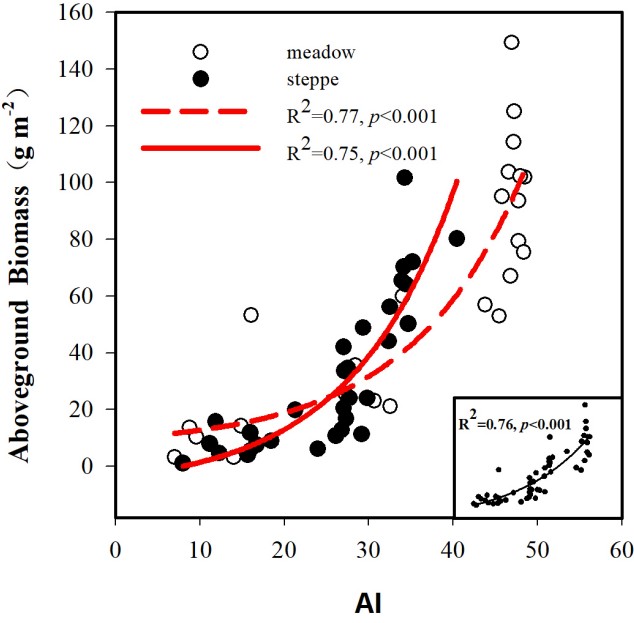

**Figure 5   Relationships between aboveground biomass and the aridity index (AI).** The black solid circle and hollow circle represent the samples collected in alpine steppe and alpine meadow respectively. The solid lines represent the fitting curves of alpine steppe, and the dotted lines represent the fitting curves of alpine meadow.

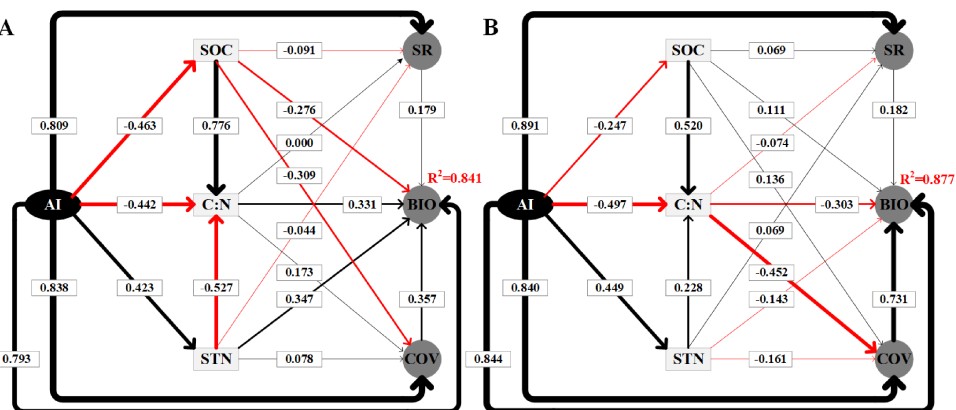

**Figure 6   Using the SEM to analyze the directly and indirectly effects among variables in alpine steppe (Graph A) and alpine meadow (Graph B), respectively.** The standardized total coefficients are listed on each significant path. The thickness of the solid arrows reflects the magnitude of the standardized SEM coefficients, the black solid line represents the positive effect while the red solid line represents the negative effect. The AI, SOC, C: N, STN, SR, BIO, COV represent aridity index, soil organic carbon, the soil carbon to nitrogen ratio, soil total nitrogen, species richness, aboveground biomass, coverage, respectively.

and temperature (*Sun, Cheng & Li, 2013*; *Sun & Qin, 2016*) in alpine grassland. Although the contributions of MAP and MAT to BIO were all higher than 57% in our experiments, the AI had the greatest contribution (72.16%) on BIO (Fig. 3). As to identifying the

complex network of causal relationships in ecosystems by SEM (*Sun, Ma & Lu, 2017*), the direct effects of the AI on plant growth and photosynthesis cannot be ignored (*Chimner et al., 2010*; *Shi et al., 2014*; *Sun, Cheng & Li, 2013*). Different levels of the AI would lead to significant differences in plant photosynthetic rate (*Liu & Chen, 1990*), and a lower AI level could reduce the stomatal opening of the leaves; as a result, the $CO_2$ supply was blocked, reducing photosynthesis (*Zhang et al., 2011*). However, at this time, plant leaves can stabilize the function of the photosynthetic mechanism by reducing the synergistic effects of light capture, heat dissipation, and enzyme activity regulation to achieve the accumulation of dry matter (*Flexas & Medrano, 2002*). When the AI was at a higher level, plant leaves can coordinate the relationship between carbon assimilation and water consumption for transpiration through stomatal conductance regulation (*Paoletti, Raddi & La Scala, 1998*), thus maintaining a high photosynthetic rate and water use efficiency and promoting plant growth (*Chengjiang & Qingliang, 2002*). As a result, if AI at a lower level, the increases in BIO is minimum; and once AI is larger than a certain point, BIO would increase rapidly (Fig. 5). But in any case, when the AI is within a certain range, there are always positive effects on aboveground biological vegetation traits (BIO, species diversity, and coverage) of the AI in both alpine steppe and alpine meadow (Fig. 6).

However, these positive effects do not equate to the same response mechanism of vegetation traits to the AI in different grassland types on the northern Tibet Plateau. In contrast, earlier studies revealed that grasslands with the different AI levels showed different response patterns to vegetation biomass (*Chen et al., 2016*). Consistent with our research, in an alpine steppe (the AI largely was less than 27.22), our experiments showed that C: N and STN had positive effects while SOC had negative effect on BIO (Fig. 6A). However, the opposite effects of soil on BIO were detected in alpine meadow (the AI largely was greater than 43.79), as C: N and STN have negative effects while SOC has positive effects on BIO (Fig. 6B). Generally, plant growth in alpine grasslands was limited by soil N (*Ladwig et al., 2012*; *Lambers et al., 2009*). Illustrated by comparing the nitrogen concentration of sub-humid and humid grasslands (0.09% for the alpine steppe and 0.18% for meadow, Table 1), STN increases with the increase in water supply (Fig. 6) due to the rainfall and biological nitrogen fixation dominated by hydrothermal conditions (*Aber et al., 1998*). However, when the water supply exceeds the needs of plant growth, the excessive water content inhibits the decomposition of organic matter, and the STN increases further by the rainfall and biological nitrogen fixation. This ecological process can be harmful to vegetation in bottom land (*Aber et al., 1998*). However, these patterns do not apply to SOC because of its lower solubility (*Hyun et al., 1998*). That is, why the mean value of SOC was largely the same (2.23% for alpine steppe and 2.37% for meadow) in two vegetation types despite gradual decline in soil organic matter from increased the AI. This is because the increasing AI will intensify the activities of soil microorganisms and soil animals and accelerate soil respiration, inevitably leading to the acceleration of the release of $CO_2$ from the soil carbon pool (*Anderson, 1992*) and resulting in the imbalance between the input and output of soil carbon (Fig. 6). In this case, soil moisture was a major limiting factor, and the drier climate of the alpine steppe produced a lack of water-soluble organic carbon that was absorbed directly by the root and soil microorganisms, constraining the development

**Table 2  Summary of the direct, indirect and total effects of variables (AI, STN, SOC, C: N, COV, SR, BIO) in the SEM of alpine steppe and alpine meadow. Effects were calculated with standardized path coefficients.**

**A   Direct Effect** (Alpine steppe)

| Variable | AI | SOC | STN | C:N | COV | SR |
|---|---|---|---|---|---|---|
| SOC | −.463* | .000 | .000 | .000 | .000 | .000 |
| STN | .423** | .000 | .000 | .000 | .000 | .000 |
| C:N | .140* | .776*** | −.527*** | .000 | .000 | .000 |
| COV | .609*** | −.444 | .170 | .173 | .000 | .000 |
| SR | .814*** | −.092 | −.043 | .000 | .000 | .000 |
| BIO | .109 | −.358* | .469** | .270 | .357** | .179 |

**B   Direct Effect** (Alpine meadow)

| Variable | AI | STN | SOC | C:N | COV | SR |
|---|---|---|---|---|---|---|
| STN | .449* | .000 | .000 | .000 | .000 | .000 |
| SOC | −.247 | .000 | .000 | .000 | .000 | .000 |
| C:N | −.471** | .228 | .520*** | .000 | .000 | .000 |
| COV | .733*** | −.058 | .371*** | −.452*** | .000 | .000 |
| SR | .842*** | .086 | .108 | −.074 | .000 | .000 |
| BIO | .105 | −.047* | −.023* | .041 | .731*** | .182 |

**C   Indirect Effect** (Alpine steppe)

| Variable | AI | SOC | STN | C:N | COV | SR |
|---|---|---|---|---|---|---|
| SOC | .000 | .000 | .000 | .000 | .000 | .000 |
| STN | .000 | .000 | .000 | .000 | .000 | .000 |
| C:N | −.582 | .000 | .000 | .000 | .000 | .000 |
| COV | .200 | .134 | −.091 | .000 | .000 | .000 |
| SR | .024 | .000 | .000 | .000 | .000 | .000 |
| BIO | .684 | .083 | −.122 | .062 | .000 | .000 |

**D   Indirect Effect** (Alpine meadow)

| Variable | AI | STN | SOC | C:N | COV | SR |
|---|---|---|---|---|---|---|
| STN | .000 | .000 | .000 | .000 | .000 | .000 |
| SOC | .000 | .000 | .000 | .000 | .000 | .000 |
| C:N | −.026 | .000 | .000 | .000 | .000 | .000 |
| COV | .107 | −.103 | −.235 | .000 | .000 | .000 |
| SR | .048 | −.017 | −.038 | .000 | .000 | .000 |
| BIO | .740 | −.096 | .134 | −.344 | .000 | .000 |

**E   Total Effect** (Alpine steppe)

| Variable | AI | SOC | STN | C:N | COV | SR |
|---|---|---|---|---|---|---|
| SOC | −.463 | .000 | .000 | .000 | .000 | .000 |
| STN | .423 | .000 | .000 | .000 | .000 | .000 |
| C:N | −.442 | .776 | −.527 | .000 | .000 | .000 |
| COV | .809 | −.309 | .078 | .173 | .000 | .000 |
| SR | .838 | −.091 | −.044 | .000 | .000 | .000 |
| BIO | .793 | −.276 | .347 | .331 | .357 | .179 |

**F   Total Effect** (Alpine meadow)

| Variable | AI | STN | SOC | C:N | COV | SR |
|---|---|---|---|---|---|---|
| STN | .449 | .000 | .000 | .000 | .000 | .000 |
| SOC | −.247 | .000 | .000 | .000 | .000 | .000 |
| C:N | −.497 | .228 | .520 | .000 | .000 | .000 |
| COV | .840 | −.161 | .136 | −.452 | .000 | .000 |
| SR | .891 | .069 | .069 | −.074 | .000 | .000 |
| BIO | .844 | −.143 | .111 | −.303 | .731 | .182 |

**Notes.**
*Correlation is significant at the 0.05 level.
**Correlation is significant at the 0.01 level.
***Correlation is significant at the 0.001 level.
AI, SOC, C: N, STN, SR, BIO, and COV represent aridity index, soil organic carbon, the soil carbon to nitrogen ratio, soil total nitrogen, species richness, aboveground biomass, and coverage, respectively.

of root systems (*Alexandre et al., 2016*; *Kick, Sauerbeck & Führ, 1964*). In contrast, in the humid environment of alpine meadow, the increase in water-soluble organic carbon in the soil led to significant improvements in plant productivity (*Atkinson, Fitzgerald & Hipps, 2010*). This indicated that the AI (water and heat conditions) affected vegetation growth not only directly but also indirectly by modifying soil properties (i.e., soil moisture and soil nitrogen mineralization and availability) (*Ruppert et al., 2012*; *Shaver et al., 2000*; *Wan et al., 2005*).

## CONCLUSIONS

In this study, we verified that the AI (integrated MAT and MAP) was the primary limiting environmental factor for vegetation traits in alpine grassland. Additionally, the different AI govern the different response patterns of BIO in the alpine steppe and the alpine meadow, respectively. The findings imply the important role of AI regulating vegetation dynamics, which need to be explored deeply in future research. Nevertheless, due to the unevenly distribution of the sampling sites throughout the vast region of northern Tibet plateau, the spatial heterogeneity of vegetation across regional environmental gradients limited this research. Hence, we should pay more attention to reduce spatial heterogeneity (i.e., optimize sampling methods or analytical methods) to verify the effect of AI on variations of biomass.

## ACKNOWLEDGEMENTS

The author thanks anonymous reviewers for providing invaluable comments on the original manuscript. The author thanks the research team for the data collection of this study, Dr. Sun and Dr. Zeng for their guidance of this paper, and his girlfriend (Ms. Zhang) for her support of his work.

### Funding

This research was funded by the National Science Foundation of China (Grant No. 41871040 and 41501057). The funders had no role in study design, data collection and analysis, decision to publish, or preparation of the manuscript.

### Grant Disclosures

The following grant information was disclosed by the authors:
National Science Foundation of China: 41871040, 41501057.

### Competing Interests

The authors declare there are no competing interests.

### Author Contributions

- Biying Liu conceived and designed the experiments, performed the experiments, analyzed the data, prepared figures and/or tables, authored or reviewed drafts of the paper.
- Jian Sun conceived and designed the experiments, performed the experiments, analyzed the data, contributed reagents/materials/analysis tools, prepared figures and/or tables, authored or reviewed drafts of the paper, approved the final draft.
- Miao Liu analyzed the data.
- Tao Zeng contributed reagents/materials/analysis tools, approved the final draft.
- Juntao Zhu contributed reagents/materials/analysis tools, authored or reviewed drafts of the paper.

## Data Availability

The raw measurements are available as a Supplemental File.

## Supplemental Information

Supplemental information for this article can be found online at http://dx.doi.org/10.7717/peerj.7272#supplemental-information.

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
