# Peer review of "The aridity index governs the variation of vegetation characteristics in alpine grassland, Northern Tibet Plateau"

_PeerJ, doi:10.7717/peerj.7272_

## Round 0.1 · original submission · Major Revisions

We have received the comments on your manuscript from two reviewers. Based on the comments of two reviewers, I advise that you further modify and improve your paper.



Reviewer 1 ·

Basic reporting

Line 37, "..was the largest" comparing to what?

Line 40-43, those sentences are not scientifically written. Better change to "soils with higher carbon to nitrogen ratio and soil total nitrogen had larger biomass...", I assume this is a correlation test.

Line 43, "the most critical factor" comparing to what other factors?

Line 61, this sentence is not an example of altered climate changing grassland ecosystem, it has changes of grassland altering climate: "grassland degradation and disappearance affect climate..." Do the authors try to illustrate the impact is bidirectional?

Line 62, should soils be one part of the alpine grassland?

Line 71, use "influence" instead of "determine"?

Line 79, what is "50 mm intervals"? authors could just give the mean annual precipitation.

Line 81, note that the increases in soil temperature is due to the increases in air temperature. I think the sentence is line 80 better to be "Responses of grassland to climate change also varied if soil properties are different"?

Line 83, again, the word "determine" is strong, use "influence" instead

Line 87, omit "In addition"

Line 89, omit "In consequence"

Line 91, past tense too

Line 92, i don't see a "cause-effect" here, omit the first half sentence and change the other half to "To explain the mechanism of environmental variables affecting biomass in different vegetation types, we compared the contributions of the critical environmental variables to vegetation biomass between alpine steppe and meadow."

Figure 5, looks like an exponential curve would be more fitted than a linear

Experimental design

Line 110, what samples? Put "Plant BIO" here

Line 121, "at 4degreeC until measuring chemical properties"

Line 123, concentration not content, right?

Line 124, missing sentences describing quality control (e.g. srm recovery, etc)

Line 142, omit "Next"

Line 147, omit "finally", this is a new paragraph, so don't need to use "finally"

Line 147, what are "other variables"? do authors mean all the independent variables?

Validity of the findings

Line 154, relationship between... and...

Line 157-158, for correlation test, it is the r (not r2) usually being reported.

Line 160, typo? "...the least effect soil on species richness..."

Line 163, omit the first half sentence, start with results

Line 170, omit "obviously"

Line 172, ", and 3.18 to 149.30..."

Line 172, "latter difference"? do you mean differences in BIO?

Line 174-175, why using 30 as a threshold here? authors can just write the exact value out

Line 179, this sentence can be written more straightforward. something like "with the same increase/decrease of AI, larger increases/decreases would be found in steppe than in meadow"

Line 186, those biotic factors are environmental factors, so cannot use "both of which" later on. And is this sentence meaning "changes in BIO (due to changes in climate) also influence the environmental variables?" I am confused

Line 194-197, are those coefficients (e.g. -0.091, 0.111) significant? if not, there is just no correlation (effects).

Line 201, why cite Figure 6 for "our hypothesis"? I don't see any relevance

Line 203, if authors have "increased or decreased" for BIO in the first half sentence, should also use "more or less precipitation.." in the other half sentence. Or simply write "the aboveground biomass/vegetation index depends on the precipitation and temperature"

Line 206, I think this is obvious because AI is an index with information of both precipitation and temperature

Line 224, this sentence is talking about earlier studies, and I was expecting to see examples from literature, but authors give their own examples instead

Line 227-288, again, why using 30 as the divided number?

Line 235, if the decomposition rate is slower, there will be less nitrogen releasing out too, why the soil total nitrogen increases instead?

Line 243, should also discussing leaching (dissolved organic carbon)

Additional comments

This paper explores the effect and contribution of environmental factors on aboveground biomass in alpine steppe and alpine meadow in Tibetan Plateau in China. Environmental factors include annual mean temperature, annual mean precipitation, aridity index, soil total nitrogen, soil total carbon, soil C:N ratio, coverage and species richness. The main finding of this study is that, the aridity index is the critical factor that influencing the aboveground biomass in alpine grassland. I believe this paper adds important value to the community.

Reviewer 2 ·

Basic reporting

1. There are some minor errors of writing in the manuscript (e.g. line 82& line102), please carefully check it.
2. In figure 4, the bottom part of the confidence interval line of the error bar was missing, please revise it.

Experimental design

1. The MAT and MAP data used form Worldclim database website in the study was not well introduced, the authors should make more description of it (deriving algorithms, quality, resolution, accuracy, et al)
2. The introduction of SEM method applied in the study was not clear, which makes it hard to comprehend how SEM works, and some more introduction should be added. Please make more explanation of the path coefficients derived from SEM, how did you define the grades of thickness of arrow in figure6?
3. As shown in DEM layer of figure1, the Tibetan Plateau is vast with great elevation changes, which result in the high spatial heterogeneity of natural conditions. The sampling points of the study was concentrated in a line located almost in the same latitude in the middle of Tibet administrative division. Is that enough to explain the phenomenon of entire plateau? Please make further explanation about that.

Validity of the findings

1. The description of conclusion was ambiguous, please revise the part in a more specific way. In addition, the prospect of further study is recommended to be added in the part.

Additional comments

This study derived that AI was the dominating factor affecting the grassland vegetation variation characteristics through correlation analysis and redundancy analysis, then discussed the influence factors of two grassland types via structural equation modelling (SEM). The sampling method was robust and the experiment was well designed. However, there should be some enhancements and supplements of the manuscript before publishing.

---

## Round 0.2 · Minor Revisions

Based on the reviewer's comments of the second round on your manuscript , I suggest that you moderately revise your paper before it be accepted for publication.

Reviewer 1 ·

Basic reporting

None

Experimental design

None

Validity of the findings

Now the the fitted line for relationship between AI and Aboveground Biomass is exponential, authors can add few sentences discussing that. For example, if AI is not reaching to a certain point (20-30), the increases in Aboveground Biomass is minimum. But once AI is larger than 20-30, aboveground biomass would increase rapidly.

Additional comments

I really appreciated authors' hardworking on revising this paper. I only have one minor suggestions mention above. I suggested an acceptance for this paper.

Reviewer 2 ·

Basic reporting

no comment

Experimental design

1. According to your explanation on the representativeness of the sampling points in Tibet Plateau in this study, it is not enough for the current samples to discuss the entire plateau. I would recommend you revise the study area in your title as central Tibet or southwestern Tibet Plateau.

Validity of the findings

no comment

Additional comments

Thank you for revising this manuscript. The revised version is a much clearer presentation of your research than the previous one. I believe that this manuscript is ready for publications after some minor modifications.

---

## Round 0.3 · accepted · Accept

After comparing your revised manuscript (version 2) with the review comments of the previous round, I have decided to accept your paper for publication.